# Rethinking the Design of Learning based Inter-Patient Registration using Deformable Supervoxels

**Mattias P Heinrich**                                                HEINRICH@IMI.UNI-LUEBECK.DE

*Institute of Medical Informatics, Universität zu Lübeck, Germany*

## Abstract

Deep learning has the potential to substantially improve inter-subject alignment for shape and atlas analysis. So far most highly accurate supervised approaches require dense manual annotations and complex multi-level architectures but may still be susceptible to label bias. We present a radically different approach for learning to estimate large deformations without expert supervision. Instead of regressing displacements, we train a 3D DeepLab network to predict automatic supervoxel segmentations. To enable consistent supervoxel labels, we use the warping field of a conventional approach and increase the accuracy by sampling multiple complementary over-segmentations. We experimentally demonstrate that 1) our deformable supervoxels are less sensitive to large initial misalignment and can combine linear and nonlinear registration and 2) using this self-supervised classification loss is more robust to noisy ground truth and leads to better convergence than direct regression as supervision. Public code and dataset: github.com/multimodallearning/slic_reg

**Keywords:** Supervoxel segmentation, inter-patient registration, noisy labels

## 1. Motivation / Related Work

While early work in DL-based registration had explored using the predictions of conventional methods as strong but noisy **displacement supervision**, more recent work focused on weak **metric- or/and label supervision**. Label supervision has great promise but may still require a robust pre-alignment, multi-scale architecture and dense set of labels to avoid local minima and structural bias. Yet non-learning methods still excel in comprehensive medical image registration benchmarks and outperform metric-supervised approaches cf. https://{curious2019},{empire10},{learn2reg}.grand-challenge.org.

We hence hypothesise that methods using displacement supervision have been outperformed by metric-supervision despite the fact that conventional (non-learning) algorithms provide better results due to the difficulty of training a deep network for voxel-wise regression, which are less robust to optimise than segmentation architectures.

**Contribution**: We propose a novel loss and fitting function for optimising deep networks for inter-patient registration based on automatically deformed and thus anatomically consistent supervoxel segmentations as training data. Deformable registration is thus solved using segmentation of a reasonably small number of supervoxel classes instead of direct regression of transformation parameters. The displacement field is then indirectly estimated through the correspondence of the predicted supervoxels.

**Related Work**: Our work is related to the idea of pairwise supervoxel classification forest (Kanavati et al., 2017), which is however based on handcrafted context features and does neither employ displacements as supervision nor deep CNNs for predictions. Similarly, (Heinrich et al., 2013) used supervoxel alignment to register CT and MRI scans and provided the concept of complementary layers of over-segmentations, but without any learning.

## 2. Method

We choose deeds github.com/mattiaspaul/deedsBCV as conventional baseline registration framework (as described in (Xu et al., 2016)), which excelled in both intra- and inter-patient registration of abdominal and thoracic CT scans. First, a single template scan is selected from which both linear and deformable transformations are estimated to all training images without any supervision. Second, the template scan is automatically over segmented using the SLIC supervoxel algorithm with subsequent post-processing to obtain an equal number (here 127 + background) of connected supervoxels within a rough body mask. This step is repeated 16 times with slightly varying initialisations so that each voxel can be identified by a 16-tuple of supervoxels (please also see (Heinrich et al., 2013) for details). By simultaneously assigning each voxel to multiple *layers* of supervoxels its spatial position in the canonical space of reference coordinates is more accurately defined. Next, the reference supervoxels are spatially transferred to all training images using deeds' displacement fields. Third, a deep CNN is trained to densely predict the over-segmentations for unseen images (see Fig. 1 for a qualitative result). We designed a compact 3D DeepLab (MobileNetV2 with ASPP) (Sandler et al., 2018) for best performance on a small dataset. That means for each voxel a total of 16 softmax scores (with 128 class probabilities) is estimated and one-to-one correspondences between reference and test image could be directly obtained. Yet, to obtain an optimal displacement field that balances the potentially diverging correspondences and avoids erroneous hard assignments, we implement another step that optimises the alignment of all supervoxels using the Adam optimiser with a diffusion regularisation penalty (details are found in source code).

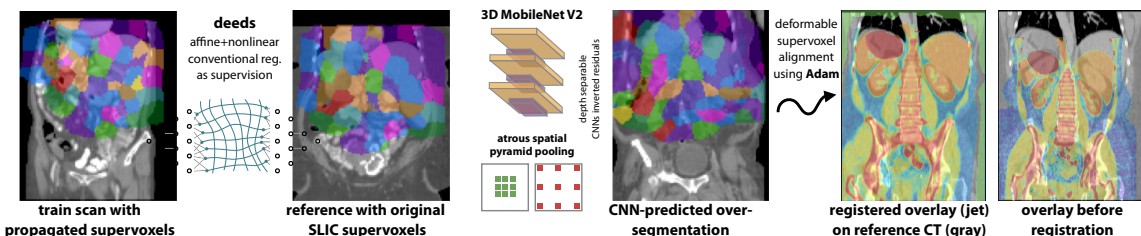

Figure 1: Slic-Reg: concept of learning registration through supervoxel segmentation.

## 3. Experiments and Results

We evaluate and compare our proposed Slic-Reg method on the Beyond the Cranial Vault multi-label abdominal CT dataset (see (Xu et al., 2016) for a description) with only limited pre-processing: resizing to 2mm isotropic resolution and cropping/padding to same dimensions using centre-mass alignment. Note that the Learn2Reg challenge used this dataset with additional affine pre-alignment making the registration much simpler than ours. 20 scans (case IDs 2,3,5,6,...) were used for training and 10 scans for test (IDs 1,4,7,...), where #38 was selected as reference template. As baseline we used PDD-Net a state-of-the-art metric supervised DL-reg method github.com/multimodallearning/pdd_net. Additionally, we employ the same backbone (3D DeepLab with MobileNetV2 + ASPP, 19 CNN lay-

Table 1: Quantitative comparison of Dice overlap of initial alignment, PDD-Net, displacement regression and our proposed **Slic-Reg**. The ruler separates methods based on backbone: 3D UNet or 3D DeepLab. In addition to four large organs, where our approach outperforms PDD by 9 points and regression by 14 points, we also evaluated a total of 13 structures. (RW is short for random walk regularisation)

| Method | spleen 🟥 | r.kidney 🟨 | l.kidney 🟩 | liver 🟪 | **avg(4)** | **avg(13)** |
|---|---|---|---|---|---|---|
| initial not registered | 18.0±17.7 | 12.5±14.9 | 9.0±12.5 | 26.2±18.1 | 16.4±16.6 | 8.8±13.4 |
| PDD-Net two warps (s.o.t.a.) | 48.8±26.5 | 49.0±23.3 | 42.3±23.0 | 60.2±23.6 | 50.1±24.0 | 29.1±25.3 |
| 3D UNet + Regression + RW | 30.2±16.8 | 41.9±17.8 | 35.3±8.8 | 57.4±8.5 | 41.2±16.7 | 23.2±18.2 |
| 3D UNet + Slic + Adam | 57.1±9.4 | 38.4±12.1 | 41.8±12.1 | 71.8±9.0 | 52.3±17.0 | 30.0±22.1 |
| 3D DeepLab + Regression + RW | 31.1±16.6 | 45.6±15.6 | 43.2±10.6 | 57.7±8.4 | 44.4±15.9 | 25.1±19.3 |
| **Slic-Reg** 3D DeepLab + Slic + Adam | 62.4±9.2 | 50.8±11.8 | 49.1±13.5 | 74.1±8.6 | **59.1±14.6** | 31.8±23.6 |

ers and 1.2M trainable parameters) but with a regression instead of a classification head (DL-regression). Note that the classification head has an additional 500k parameters and requires memory checkpointing to be trained with limited GPU-RAM (8 GByte). The regression target is computed only at supervoxel locations (1024 per scan), which helped to avoid overfitting and affine augmentation ($\sigma = 0.07$) was used for all methods. The networks were each trained for 5000 iterations with mini-batch size =4 and a learning rate of 0.004 (Adam) with mixed precision (further hyperparameters are found in our public code).

## 4. Discussion and Outlook

We have proposed a completely new concept for DL-based inter-patient registration that replaces displacement regression with supervoxel segmentation. While using the same conventional algorithm as noisy supervision we could show that substantially improved accuracies are achieved on a challenging abdominal dataset. We also outperformed PDD-Net the best ranking metric-supervised approach of the Learn2Reg 2020 challenge. Future work would extend this concept to multi-modal and pair-wise registration.

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
