# OpenReview forum: "Rethinking the Design of Learning based Inter-Patient Registration using Deformable Supervoxels "
_MIDL.io/2021/Conference/Short — MIDL 2021 Poster_

### Official Review · Reviewer_Ge6k · 2021-04-29

**Confidence:** 5
**Final Rating:** 4

**Summary:**

The paper introduces the idea of utilizing supervoxel predictions from deep neural networks for medical image registration. The method is evaluated on a publically available challenge dataset developed for multi-atlas image registration in abdominal CT. The paper is well written and clearly presented.

**Strengths:**

The approach seems novel.
Results on a challenging abdominal CT registration dataset are promising compared to state-of-the-art approaches.
The source code has been made available, which is a bonus.

**Weaknesses:**

Only the performance on large organs (spleen, kidneys, liver) is shown in detail. More challenging and "mobile" organs like the pancreas are presumingly hidden in the average score of 13 structures.
Only one reference template was utilized. It would be interesting to see the performance of the approach when used in a multi-atlas label fusion setting.

**Deanonymize Review:**

no

**Detailed Comments:**

The Dice score performance seems still low compared to modern automated multi-organ segmentation networks (3D U-Net) trained for the segmentation task. It would be interesting to compare how the registration approach compares to a CNN trained on only one example or N examples when comparing to a multi-atlas label fusion approach. See https://arxiv.org/abs/1711.06853 for the performance of a CNN on the same dataset.

**Justification Of The Rating:**

An interesting piece of work that worth discussing at the conference. It might open up some interesting research directions in the field of medical image registration, especially for more challenging inter-patient and multi-modality registration tasks.

**Paper Type:**

methodological development

**Special Issue:**

yes

---

### Official Review · Reviewer_GxCu · 2021-04-30

**Confidence:** 3
**Final Rating:** 4

**Summary:**

This paper proposes a novel, so-called Slic-Reg approach for DL-based inter-patient image registration without the need for expert supervision.

Instead of solving a multivariate regression task, the author formulates the deformable registration as a two-fold problem based on supervoxels. As such, the underlying estimation of linear and nonlinear transformations is transferred to a classification problem (the inference of the supervoxel labels with a 3D DeepLab model) combined with a subsequent, indirect estimation of the displacement field in the lower dimensional, supervoxel space.


**Strengths:**

The paper is clearly written and structured. Also, source code and data are available, which is very helpful to understand how the key features of the proposed method interact with each other. Despite the limited available page number, the paper includes a comprehensive evaluation against other state-of-the-art registrations based on the example of multi-label abdominal CT data.

In terms of future analysis, it would be interesting to see how sensitive the framework is with respect to the baseline registration method used to generate the training data and to the choice of hyperparameters, such as the number subvoxel classes and the number of iterations/initializations.


**Weaknesses:**

Although the paper is well written, the introduction of the methodological framework would benefit from a less strict page limit. A more detailed explanation of how the individual features, i.e. the 3D DeepLab and the subsequent Adam optimization, interact with each other and how the exact data flow looks like at training/testing time, could further improve the quality of the paper. Given the strict page limit, I understand that it is difficult to find a good balance between methods, results and discussion though.

**Deanonymize Review:**

no

**Justification Of The Rating:**

I don't see any major weaknesses of this paper. It proposes a sound methodology, presents a comprehensive evaluation and fits the scope of MIDL.  For these reasons, I recommend acceptance too learn more about the method during the conference.

**Paper Type:**

methodological development

**Special Issue:**

no

---

### Meta-Review · Program_Chairs · 2021-05-09

**Recommendation:** Accept (Poster)
**Confidence:** 5

**Metareview:**

Very nice work with strong support from both reviewers.

---

### Decision · Program_Chairs · 2021-05-11

Accept (Poster)